# TAME trial: a multi-arm phase II randomised trial of four novel interventions for malnutrition enteropathy in Zambia and Zimbabwe - a study protocol

Paul Kelly [1,2,3] Lauren Bell,[3] Beatrice Amadi,[2] Mutsa Bwakura-Dangarembizi,[4,5] Kelley VanBuskirk,[2] Kanta Chandwe,[2] Miyoba Chipunza,[2] Deophine Ngosa,[2] Nivea Chulu,[2] Susan Hill,[6] Simon Murch,[7] Raymond Playford,[1] Andrew Prendergast[1,5]

For numbered affiliations see end of article.

**Correspondence to**
Dr Paul Kelly;
m.p.kelly@qmul.ac.uk

## ABSTRACT

**Introduction** Severe acute malnutrition (SAM) in children in many countries still carries unacceptably high mortality, especially when complicated by secondary infection or metabolic derangements. New therapies are urgently needed and we have identified mucosal healing in the intestine as a potential target for novel treatment approaches.

**Methods and analysis** The TAME trial (Therapeutic Approaches for Malnutrition Enteropathy) will evaluate four novel treatments in an efficient multi-arm single-blind phase II design. In three hospitals in Zambia and Zimbabwe, 225 children with SAM will be randomised to one of these treatments or to standard care, once their inpatient treatment has reached the point of transition from stabilisation to increased nutritional intake. The four interventions are budesonide, bovine colostrum or N-acetyl glucosamine given orally or via nasogastric tube, or teduglutide given by subcutaneous injection. The primary endpoint will be a composite score of faecal inflammatory markers, and a range of secondary endpoints include clinical and laboratory endpoints. Treatments will be given daily for 14 days, and evaluation of the major endpoints will be at 14 to 18 days, with a final clinical evaluation at 28 days. In a subset of children in Zambia, endoscopic biopsies will be used to evaluate the effect of interventions in detail.

**Ethics and dissemination** The study has been approved by the University of Zambia Biomedical Research Ethics Committee (006-09-17, dated 9th July, 2018), and the Joint Research Ethics Committee of the University of Zimbabwe (24th July, 2019). Caregivers will provide written informed consent for each participant. Findings will be disseminated through peer-reviewed journals, conference presentations and to caregivers at face-to-face meetings.

**Trial registration number** NCT03716115; Pre-results.

## Strengths and limitations of this study

► Efficient multi-arm trial design.
► Evaluation of potential efficacy of four novel interventions for a life-threatening disorder of great importance for global health.
► Measurement of a broad range of biomarkers and measures of mucosal pathology.
► Only 14 days of treatment will be tested; although longer treatment may be effective, the greatest need is for treatments which confer benefit in the period of highest mortality.
► There is no consensus on optimal markers of mucosal healing, hence the need for a wide range of endpoints.

## BACKGROUND AND RATIONALE

Nutritional disorders are glaring examples of health inequalities between high- and low-income countries, and within low-income countries. Malnutrition underlies almost half of all child deaths globally and therefore contributes enormously to the unacceptably high under-five mortality rates in these regions.[1] Chronic undernutrition, which usually manifests as stunting (poor linear growth), affects 40% and 27% of children in Zambia and Zimbabwe, respectively,[2] and is associated with increased mortality,[3] reduced neurodevelopmental potential and decreased long-term economic productivity.[4] Acute malnutrition usually manifests as wasting (loss of tissue), with or without oedema and is the most conspicuous of all nutritional disorders. Severe acute malnutrition (SAM) carries the highest mortality,[5] particularly if associated with medical complications. HIV has changed the epidemiology, pathogenesis and clinical presentation of SAM, and children with both conditions have a particularly high mortality.[6]

Over the last two decades, three key developments in the approach to treatment have improved the outcomes of SAM: standardised management protocols, ready-to-use therapeutic food (RUTF) and community management of acute malnutrition, which enables identification and management of children without medical complications of SAM.[7 8] However, severely malnourished children with medical complications still require hospitalisation,[9] and often fail to respond to treatment.[10] Inpatient mortality among children with complicated SAM remains up to 38%.[5 11–13] Even after discharge, children have a poor prognosis, with up to 42% mortality documented over the subsequent year.[6] In our experience, it is a subgroup of children with SAM and acute or persistent diarrhoea who pose the most difficult management challenges, although the vast majority of children with SAM and/or stunting have a degree of enteropathy.[11 14 15]

Recent studies[16 17] have taught us a great deal about the contribution of enteric infections to dysfunction of the small intestine in malnourished children. A high pathogen burden causes damage to the mucosa which exacerbates nutritional impairment and leads to further susceptibility to infection and impaired epithelial regeneration, in a cyclical process first described in the 1970s.[18 19] This mucosal damage in SAM we refer to as *malnutrition enteropathy*, which is characterised by multiple epithelial breaches, microbial translocation from the gut lumen to the systemic circulation and systemic inflammation.[20] Epithelial breaches are present in histological sections, seen in haematoxylin/eosin-stained sections and using immunofluorescence for claudin 4 and E-cadherin.[20 21] In parallel studies in adults, we demonstrated that these lesions occur in vivo using confocal laser endomicroscopy.[21] We have also identified a consistent pattern of blunted epithelial repair, with reduced glucagon-like peptide 2 (GLP2) in serum, reduced trefoil factor 3 in duodenal aspirates[21] and a strong transcriptomic signature of impaired mucosal defence. We also observed two further immunological abnormalities: low-level false-positive coeliac-like autoantibodies,[20] and upregulation of SMAD7 somewhat similar to the pattern seen in Crohn's disease.[22] We have previously reported that markers of microbial translocation and systemic inflammation were dramatically increased in children with SAM in Zambia compared with healthy controls.[20] Together, these abnormalities indicate there is substantial structural and functional damage to the small intestine,[12] and that this primary gut pathology is associated with systemic sequelae. Importantly, biomarkers of these processes have been associated with mortality among children hospitalised with complicated SAM.[12]

Current treatment guidelines for SAM are not well supported by an evidence base, and there is a dearth of clinical trial data; in particular, there are no specific interventions to target enteropathy in SAM.[5] In a systematic review,[23] only three trials were found which inform management of SAM and persistent diarrhoea, and no trials dealing with the HIV-infected child. We therefore believe that novel therapeutic approaches are urgently needed, and that a series of small phase II trials will help guide development of a new generation of treatments. These trials should focus on repairing damage to the small intestinal mucosa, as we now have substantial evidence that this plays a central role in the genesis of systemic inflammation, bacterial translocation and sepsis, though it is possible that it could be a consequence of inflammation. We propose evaluation of four new interventions: bovine colostrum, N-acetyl glucosamine, teduglutide and budesonide.

## OBJECTIVES

To determine if four new interventions (bovine colostrum, N-acetyl glucosamine, teduglutide and budesonide) can ameliorate malnutrition enteropathy in children with severe acute malnutrition in Zambia or Zimbabwe. It is hoped that one or more of these interventions may be sufficiently promising to take through into phase III trials.

## METHODS AND ANALYSIS
### Trial design

There is emerging awareness that multi-arm phase II clinical trials offer a more efficient approach to identification of new therapies than parallel conventional trial designs, in which one agent is evaluated against a control. Our recent work on the pathophysiology of malnutrition enteropathy suggests that mucosal healing may be central to reducing mortality, but we do not yet know the efficacy of several plausible therapeutic approaches. In order to evaluate these four new potential therapies efficiently and quickly, they will be compared side-by-side using biomarkers of pathophysiology as endpoints. This design has the following advantages:

1. The four novel therapies will each be evaluated against standard care;
2. Allocation to all five treatment groups will run in parallel, so that all children will be randomised to any of the five groups at any point in time to avoid biasses due to seasonal variation;
3. The trial endpoints will evaluate a range of biomarkers that capture different domains of malnutrition enteropathy, to allow a comprehensive non-invasive assessment of mucosal healing;
4. A subgroup of children (those recruited in Lusaka) will have small intestinal endoscopy in a unit with a track record of safety in children;
5. Endoscopic biopsy will provide evaluation of healing and ensure that the selected therapy does not induce unwanted immune or other effects;
6. The overall trial design provides an efficient way of identifying one or more candidates for a future phase III trial.

### Study setting and participant recruitment

The study will be conducted in hospitals in Lusaka, Zambia, and Harare, Zimbabwe, which have experience

of several previous studies of severe acute malnutrition.[11 20 24] Children hospitalised with SAM will be eligible for inclusion if they are aged 6 to 59 months, of either sex, are an inpatient in one of the children's wards of the participating hospitals, have initiated transition (from F75 feed to F100 or RUTF), and clinically stable. SAM is defined using WHO definitions: weight-for-length/height Z-score of less than −3, or mid upper arm circumference of less than 11.5 cm and/or bilateral pedal oedema. During the Stabilisation Phase of treatment of SAM, readiness for transition is determined by the ability to finish F75 feed by mouth and reduction in oedema (if patient was admitted with oedematous SAM). Such a child is deemed to be 'clinically stable' and in general is alert, not requiring oxygen, not hypothermic, hypoglycaemic and is not in shock or dehydrated.

Children will be excluded if they have weight less than 5 kg, a neurological disability or oro-facial abnormality which would explain or partly explain poor feeding, haemoglobin concentration 6 g/dL or less, contraindication to any of the trial treatments (eg, allergy to cow's milk protein) or any underlying condition, other than HIV, which in the opinion of the investigator would put the subject at undue risk of failing study completion or would interfere with analysis of study results. If the caregiver is unwilling to remain in hospital for the duration of the study treatment (14 days) they will not be recruited.

### Interventions: investigational therapies

Colostrum is the first liquid secreted by the lactating breast, and breastfed children take it for approximately the first 3 days of life. It is similar to breast milk, but with higher protein content. Bovine colostrum is available as a high-protein bovine colostrum powder (Neovite) for use as a health-promoting nutritional supplement. Colostrum contains nutrients, immunoglobulins and growth factors, including epidermal growth factor and insulin-like growth factor 1 (IGF-1); it has been shown to reduce intestinal permeability in adults.[25] Bovine colostrum is potentially allergenic in theory, though there is no evidence that cow's milk products are harmful in children with SAM and the standard therapeutic feeds (F75 and F100 recommended by WHO) contain milk proteins.

N-acetyl glucosamine (GlcNAc) is a natural aminosugar present on every cell surface. All breastfed children consume GlcNAc in human milk throughout lactation.[26] Impaired glycosylation of glycosaminoglycans has been noted for many years in oedematous malnutrition, with reduced concentrations of glycosaminoglycans in blood, urine, kidney, brain and small intestine. Specific consequences of reduced heparan sulfate expression include gut epithelial leakiness with hypoalbuminaemia.[27] GlcNAc administration has been demonstrated to restore the intestinal epithelial charged barrier in Crohn's disease.[28] As with all sugars, GlcNAc may theoretically induce osmotic diarrhoea if not absorbed in the small intestine. Although this has not been observed clinically

in animals or in older children, we will use a dose escalation schedule to minimise the chance of this happening.

Teduglutide is a long-acting form of GLP2 which has proven efficacy in intestinal failure, improving absorption and reducing the need for parenteral support.[29] GLP2 is a hormone secreted by L cells in the terminal ileum, which drives epithelial repair and mediates intestinal adaptation by increased cellular proliferation and villus hypertrophy. Teduglutide will be given by subcutaneous injection (0.05 mg/kg/day) daily for 14 days. In a recently published description of tolerability of 12 weeks of teduglutide in children in the UK and the USA,[30] vomiting was attributed to teduglutide in 10% of recipients and fever in 14%. No children had treatment withdrawal because of adverse events. Three adverse events were considered of special interest (intestinal obstruction, fluid overload and biliary derangements) but none were observed.

Budesonide is a corticosteroid which reduces inflammation in the gut but is then rapidly broken down in the liver which minimises systemic effects. Budesonide is standard therapy for Crohn's disease, and can be used for refractory coeliac disease. Since malnutrition enteropathy is characterised by intestinal inflammation, with infiltration of activated T cells,[31] an anti-inflammatory approach is rational. A prior trial of mesalazine in Kenya confirmed that an immuno-modulatory approach with mesalazine was safe in the setting of SAM,[32] but targeting the small, rather than large, intestine with a more potent agent may be more effective. Budesonide may cause immunosuppression and other corticosteroid effects (oedema, hypertension, glucose intolerance, osteoporosis), but usually only after longer-term administration. It is the corticosteroid of choice for intestinal disorders because it causes fewer adverse events than prednisolone because of low systemic absorption and first-pass clearance by the liver. The dose will be tapered (1 mg orally three times daily for 7 days, then 1 mg twice daily for 4 days, then 0.5 mg twice daily for 3 days) to mitigate any possible effects of adrenal suppression.

### Trial procedures

The trial schema is shown in figure 1. Children hospitalised with SAM will be enrolled from any of three hospitals (Lusaka Children's Hospital, Lusaka, Zambia; Parirenyatwa Hospital, Harare, Zimbabwe; Harare Central Hospital, Harare, Zimbabwe) once they have completed the stabilisation phase of nutritional rehabilitation and are at the point of transition from F75 therapeutic milk (low-calorie feed) to RUTF or F100 milk (feeds richer in calories). Children will be randomised in an allocation ratio of (1:1:1:1:1) to either colostrum, GlcNAc, teduglutide, budesonide or Standard Care and followed for 14 days in hospital (table 1). All children will receive Standard Care according to WHO guidelines,[9] and treatment with trial medications will commence after transition.

Randomisation codes will be prepared in advance by the trial statistician, and treatment allocation will be revealed after enrolment by opening sealed envelopes

Trial flow diagram

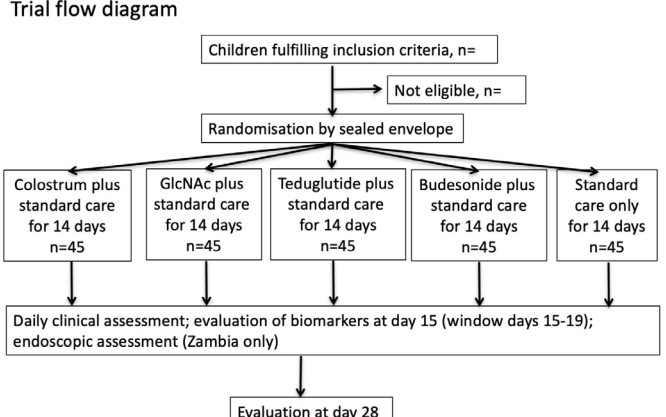

**Figure 1** Flowchart of recruitment into the TAME trial. GlcNAc,N-acetyl glucosamine; TAME, Therapeutic Approaches for Malnutrition Enteropathy.

held by the trial coordinators in each of the three study sites. Randomisation codes will be generated in permuted blocks and stratified by study site. Varying block sizes will be used; block size and distribution will not be disclosed to the study clinical teams. There will be no attempt at blinding as the interventions are readily distinguishable (eg, teduglutide is given as subcutaneous injection, so to mask all interventions would require administration of several placebo formulations). However, the primary endpoints are biomarkers so laboratory staff and data analysts will be blinded to randomised group.

### Outcomes and endpoints
The primary endpoint for this trial will be measured on day 15 (allowable window 15 to 19 days) after initiating treatment by analysis of faecal biomarkers. Gut inflammation will be measured as a composite score of faecal myeloperoxidase, neopterin and alpha-1-antitrypsin, as described below. Secondary endpoints (table 2) will be measured by daily clinical review (for clinical endpoints) during the intervention period and by collection of biological specimens on day 15 (allowable window 15 to

19 days) after initiating treatment. As endoscopic biopsies will be collected from trial participants in Lusaka, we anticipate that at least 20 biopsy scores will be available for each trial arm. Few trials have used histological endpoints in SAM, so this component of the trial is comparatively novel and therefore exploratory.

### Sample size and data analysis
Kosek *et al*[33] examined established markers of intestinal inflammation which can be used to obtain population-based measures of intestinal injury and altered intestinal function. They developed an environmental enteropathy (EE) activity score which is a composite score of three stool markers for mucosal inflammation. The composite score is derived from measurements of faecal myeloperoxidase (MPO), neopterin (NEO) and alpha-1-antitrypsin (AAT). These are stable compounds, relatively resistant to proteolysis and therefore suitable as biomarkers.

The weighted EE score is given in Equation 1:
Equation 1

$$EE\ score = 2 \times (AAT\ category) + \times (MPO\ category) + 2(1 \times NEO\ category)$$

We will use the principle of a weighted enteropathy score as our primary outcome. We will not trichotomise the variables as Kosek *et al*[33] did, but keep the score as a continuous composite variable. This is because we are interested in gauging the potential magnitude of the effect these treatments have, and applying a cut-off value to a continuous composite score could mask or discard information.[34 35] The modified Enteropathy score is given in Equation 2, with modification of weighting of units to bring measurements into a comparable arithmetical range:

Equation 2

$$EE\ biomarker\ score = 2 \times (AAT\ in\ mg/g) + 0.2 \times (MPO\ in\ \mu g/ml) + 1 \times (neopterin\ in\ \mu mol/l)$$

| Table 1 | Trial interventions in the five randomisation groups | | | | |
|---|---|---|---|---|---|
| **Group** | **C** | **N** | **T** | **B** | **S** |
| IMP | Colostrum | N-acetyl glucosamine | Teduglutide | Budesonide | Standard care |
| Presentation | Powder in capsules | Powder in capsules | Ampoules | Liquid | – |
| Number of patients | 45 | 45 | 45 | 45 | 45 |
| Dose | | | | | |
| days 1–7 | 1.5 g three times a day | 300 mg three times a day | 0.05 mg/kg daily | 1 mg three times a day | – |
| days 8–11 days 12–14 | 1.5 g three times a day 1.5 g three times a day | 600 mg three times a day 600 mg three times a day | 0.05 mg/kg daily 0.05 mg/kg daily | 1 mg two times per day 0.5 mg two times per day | – |

All IMPs are to be given for 14 days.
IMP, Investigational product.

**Table 2** Endpoints to be measured in the TAME trial

|  | Units | How measured |
|---|---|---|
| **Primary endpoint** | | |
| Faecal α-1-antitrypsin, myeloperoxidase and neopterin concentrations | mg/g, µg/ml, µmol/l | ELISA |
| **Secondary endpoints** | | |
| Lactulose:rhamnose ratio (Zambia only) | | HPLC |
| Plasma intestinal fatty acid binding protein concentration | ng/ml | ELISA |
| Plasma lipopolysaccharide (LPS) | EU/ml | Limulus amoebacyte lysate |
| Plasma LPS binding protein | ng/ml | ELISA |
| Plasma soluble CD14 and CD163 | µg/ml | ELISA |
| Plasma C-reactive protein | mg/l | ELISA |
| Plasma albumin | g/l | ELISA |
| Mortality by day 14 and day 28 | | Clinical assessment |
| Change in anthropometric measurements (weight, mid upper arm circumference, weight-for-height, MUAC) between baseline and day 14, and baseline and day 28 | | Weight and length measurements |
| Resolution of oedema between baseline and day 14 | | Clinical assessment |
| Adverse events between baseline and day 14 | | Clinical assessment |
| Serious adverse events between baseline and day 14 | | Clinical assessment |
| Days with diarrhoea (three or more loose or watery stools in 24 hours) between baseline and day 14 | | Clinical assessment |
| Days with fever (two or more recordings of core temperature of 37.8°C or higher in any 24 hours period) between baseline and day 14 | | Clinical assessment |
| Hormones: GLP-2, IGF-1 and IGFBP3 | ng/ml | ELISA |
| Morphometric measures on biopsy specimens collected in Zambia only: villus height, crypt depth, villus width, epithelial surface perimeter | µm | Microscopy |
| Mucosal inflammatory scores | Ordinal scale | Microscopy |

GLP-2, glucagon-like peptide 2; IGF, insulin-like growth factor; IGFBP, IGF binding protein; MUAC, mid upper arm circumference; TAME, Therapeutic Approaches for Malnutrition Enteropathy.

We will have a control arm, so can understand the effect of the treatment against a reference standard. The analysis will be per-protocol; no interim analysis is planned.

There are no available data from children with SAM to draw on for an understanding of the baseline variance of the EE biomarker score. The primary outcome response from each patient is the EE biomarker score at 15 days (window 15 to 19 days) after treatment initiation, adjusted for baseline values. We have assumed that the EE biomarker score is normally distributed with a common SD. To determine the sample size, we drew on two methods to establish an effect size. We assumed the difference would be larger than the likely inherent imprecision in the measurements of each outcome, and we have defined the magnitude of the effect on a standardised scale. To detect a medium/large effect of Cohen's $d$ effect size of 0.6, with 90% power and 90% confidence and a conservative correlation between baseline and follow-up estimate of 0.5, we will need a sample size of 36 per group across five groups to analyse with the analysis of covariance (ANCOVA) method (180 in total to be treated). We expect there to be approximately 10% loss to follow-up due to deaths, and 10% loss to follow-up due to other reasons, such as drug intolerance, withdrawal and missing specimens. Adjusting our sample size of 180 for 20% loss to follow-up, we therefore aim to randomise 225 patients in total (45 in each group).

We will use a mixed effect ANCOVA model to compare the environmental enteropathy activity score and secondary endpoints in each group against the control group, adjusting for several core covariates: sex (male/female), oedema (yes/no), HIV status (yes/no), diarrhoea (yes/no), breastfeeding (yes/no), baseline Weight for length z score (WLZ) scores (continuous), baseline biomarker/histology scores (continuous) and trial site. Treatment effects will be deemed statistically significant if the p value is less than or equal to 0.10 when compared with the control arm, for all four treatments. This less stringent cut-off has been chosen to reduce the likelihood of rejecting a potentially valuable treatment which might show benefit in a phase III trial. All trial committee members (see below) will contribute to a discussion at the end of the trial as to which, if any, of these treatment effects are most clinically significant.

We will not undertake any adjustments of the false-positive (type I) error rate, as the aim of this trial is to inform the treatment development process for this population, which would lead to a phase III trial if a degree of efficacy is observed. The general consensus is that adjustment for the type I error rate is not required in exploratory multi-arm multi-stage trials in phase II within the treatment development framework.[36]

If patients are discharged from hospital sooner than day 14, we will measure the primary endpoints as close as the discharge date as possible, and carry this observation forward as the primary outcome. Children who cannot submit a stool sample will be excluded from the primary analysis as the primary endpoint requires this.

### Safety reporting

All adverse events will be reviewed for causality, expectedness and severity. Serious adverse events will be reported urgently to the Trial Management Group for evaluation, and all serious adverse reactions will be regarded as unexpected (ie, SUSARs). There are two adverse events of special interest for teduglutide: fluid overload and intestinal obstruction, as these have been reported in adults on long-term teduglutide. Complicated SAM has an inpatient mortality rate of around 15%, so we anticipate up to 45 deaths in the trial; however, all deaths, regardless of causality, will be reported to the local ethics committees in Zimbabwe and Zambia according to local reporting policies. The trial will be supervised by three committees: the Trial Management Group (TMG), the Data Monitoring and Ethics Committee (DMEC, which will meet every 6 months) and the Trial Steering Committee (TSC). The TMG and TSC will review the pattern of any Serious Adverse Events (SAEs) and Serious Unexpected Adverse Reactions (SUSARs) in real time, and are empowered to introduce urgent special measures if the profile of adverse events suggests this is required. The TSC and DMEC are composed of statisticians and paediatricians, and independent of the TMG.

### Patient and public involvement

There was no patient or public involvement in the design of this trial.

## ETHICS AND DISSEMINATION
### Consent

Trial nursing and medical staff will identify eligible children whose primary caregivers will then be approached to begin the process of discussing the trial and the written, translated, information sheet (see online supplementary material) in the most appropriate language. The consent process will include honest discussions of risks and benefits of the interventions, the concept of randomisation and the purpose and intended use of samples collected, which is an important issue in this trial as the primary endpoints are all laboratory biomarkers. The consent process will continue throughout the trial, but all parents/guardians will sign to record that they have given informed consent to their child's participation, before any study procedures commence. Parents of children who cannot read will have the information sheet read out to them in the presence of family members before consent is accepted and recorded using a thumbprint. We have pioneered the use of participant visits to laboratories to facilitate valid consent.[37] This opportunity will be afforded to these parents and guardians. At least 24 hours will be available for consideration of consent; in practice, 2 to 3 days is the norm in Zambia and Zimbabwe because mothers/carers usually need to discuss enrolment with other family members. The window of time available for recruitment at and after transition allows for this consultation process.

### Dissemination of results

Dissemination of results to participants' families will take place through caregiver results meetings. We will disseminate results within the academic institutions in which the trial will be conducted and to national health research authorities. Trial results will be published following Consolidated Standards of Reporting Trials guidelines, and participant-level data set made available through a data repository. Improvements in care will ultimately occur through change in evidence-based practice. By collaborating with colleagues on design of a future interventional trial based on our results, and by sharing specimens with other research groups, we intend to push forward the research agenda for children with SAM.[38] We will also use this trial as an opportunity to increase awareness in the UK of the complexity of malnutrition. The Centre of the Cell (www.centreofthecell.org) at Queen Mary University of London is a unique, state-of-the-art science education centre with extensive experience of translating complex medical research into innovative science communication tools. We are currently working with Centre of the Cell to develop a game to explain the causes of undernutrition, including the role of the gut in nutrition, with endoscopy footage, animations and interactive gameplay . The game will be reformatted for tablets for use in Harare and Lusaka. The production of multiple formats of the tool will facilitate adults and children in the UK and Africa to access this resource and learn about current research on SAM.

### Time frame and study status

The trial has been granted ethical approval by all required regulatory authorities in Zambia and Zimbabwe. The trial should be completed in 2021.

## DISCUSSION

In the belief that reducing mortality from SAM will require novel therapeutic approaches, and recognising abundant evidence of impaired intestinal function, we have identified mucosal healing in the small intestine as a target for intervention. The TAME trial will employ a multi-arm design to identify one or more of four interventions for

taking forward into phase III trials. The strengths of this trial include an efficient design, which will evaluate four novel interventions against standard care, and very extensive assessment of mucosal healing and biomarkers to provide the greatest chance of observing potential benefit. Limitations of the trial include the short duration of treatment, ongoing uncertainty about the optimal selection of biomarkers for accurate representation of intestinal absorptive and barrier function (calprotectin is not included in the primary endpoint, for example), and the complexity of data interpretation in this group of children with serious morbidity and high predicted mortality. To ease with identification of adverse events we considered it prudent to delay recruitment until the child has stabilised. However this also means that an important, possibly critical, window for therapy during the first few days of admission may be missed. Hopefully this can be addressed in future studies.

We have considered carefully the issue of duration of treatment. Inpatient treatment of SAM lasts usually between 1 to 2 weeks and it is in this narrow time window that most, though not all,[6] of the mortality occurs. If novel therapies for reduction of mortality in SAM are to be useful at scale, they must confer benefit quickly, during the 1 to 2 week period immediately after admission. While it would be of great interest to evaluate more extended periods of therapy, the most urgent need is for new treatments in this short period of maximum morbidity and mortality. In future trials of any promising interventions, the timing of introduction of novel therapies would be an explicit research question, as they may also be of value earlier in treatment, or continued for a longer duration, including after discharge. However, for this phase II trial, we will only enrol children once they are clinically stable, at the transition phase, and provide treatment over 14 days during the typical period of hospitalisation.

Two of these agents (bovine colostrum and N-acetyl glucosamine) are nutraceuticals, food-derived and generally regarded as safe. While this is very reassuring about the low probability of adverse events, it is hard to quantify that probability with any accuracy. Regulatory agencies are more familiar with Investigational Medicinal Products (IMPs) which have well-described adverse event profiles based on careful phase I trials, so paradoxically the level of scrutiny of any potential adverse events in the TAME trial will be higher than if full Summary of Product Characteristics (SmPCs) were publically available, despite the expected low risk of significant adverse events.

Teduglutide has not previously been used for treating enteropathy in malnutrition, but there is growing evidence of its benefit in intestinal failure.[29] Children with SAM have evidence of a hyperplastic enteropathy which is likely to lead to impaired absorption of nutrients. Teduglutide is a GLP2 analogue which can enhance intestinal adaptation and reduce requirements for parenteral nutrition in patient receiving it. It may overcome some of the impaired absorption in severe enteropathies, which may enhance nutrient uptake in SAM. However there

are possible adverse effects, including increased salt and water retention which (although not reported in previous studies in children)[30] will be an adverse event of special interest in TAME. Intestinal obstruction has also been reported in adults as a consequence of mucosal hypertrophy; it too will be an adverse event of special interest.

In summary, we propose a phase II trial designed to target malnutrition enteropathy, which we believe is central to the pathogenesis of SAM, prevents recovery and is not currently addressed by current interventions. Identification of promising novel treatment approaches would provide a strong rationale for a larger efficacy trial to reduce the unacceptably high morbidity, mortality and relapse among children with complicated SAM.

**Author affiliations**
[1]Barts and The London School of Medicine, Queen Mary University of London, London, UK
[2]University of Zambia School of Medicine, Lusaka, Zambia
[3]London School of Hygiene and Tropical Medicine, London, UK
[4]Department of Paediatrics and Child Health, College of Health Sciences, University of Zimbabwe, Harare, Zimbabwe
[5]Zvitambo Institute for Maternal and Child Health Research, Harare, Zimbabwe
[6]Great Ormond Street Hospital, London, UK
[7]University Hospital Coventry & Warwickshire, Coventry, UK

**Contributors** PK, AJP, BA and MB-D originated the trial idea and obtained funding. KvB and LB provided statistical expertise. MC, DN, NC and MC collected preliminary data for sample size calculations. KC, RP, SM and SH provided specific input into design of specific elements of the trial, particularly the principal interventions. All authors contributed to and approved the final manuscript.

**Funding** The TAME trial is funded by the Medical Research Council (MR/P024033/1). AJP and MB-D are funded by the Wellcome Trust (grants 093768/Z/15/Z and 107634/Z/15/Z, respectively). These funding agencies have had no role in the preparation of this manuscript.

**Competing interests** None declared.

**Patient consent for publication** Not required.

**Ethics approval** Ethical approval for the trial has already been obtained (as of 18th March, 2019) from the University of Zambia Biomedical Research Ethics Committee (reference 006-09-17) and the Joint research Ethics Committee of the University of Zimbabwe and the College of Health Sciences (JREC/66/19). Approvals have also been obtained from the Zambian Medicines Regulatory Authority (CT/082/18) and the Medicines Control Authority of Zimbabwe (CT/176/2019).

**Provenance and peer review** Not commissioned; externally peer reviewed.

**ORCID iD**
Paul Kelly http://orcid.org/0000-0003-0844-6448

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
