## [Reviewer comments · BMJ Open]

ARTICLE DETAILS

TITLE (PROVISIONAL)	The TAME trial: a multi-arm phase 2 randomised trial of four novel interventions for malnutrition enteropathy in Zambia and Zimbabwe; a study protocol
AUTHORS	Kelly, Paul; Bell, Lauren; Amadi, Beatrice; Bwakura, Mutsa; VanBuskirk, Kelley; Chandwe, Kanta; Chipunza, Miyoba; Ngosa, Deophine; Chulu, Nivea; Hill, Susan; Murch, Simon; Playford, Raymond; Prendergast, Andrew

VERSION 1 – REVIEW

REVIEWER	Mark Manary Washington University School of Medicine
REVIEW RETURNED	15-Nov-2018

GENERAL COMMENTS	Please elaborate more about the risks and effectiveness about teduglutide, This seems like the only potential risk in the study.
--

REVIEWER	Wieger Voskuil Global Child Health Group, Academic Medical Center, Emma Children's hospital, Amsterdam, The Netherlands
REVIEW RETURNED	19-Nov-2018

GENERAL COMMENTS	Dear Authors, November 19th, 2018 Congratulations on such an important study on improving treatment outcomes in SAM children. As you rightly state; there is a massive lack of such trials. With the proposed design I do think that relatively quickly you will be able to see evidence of an effect, no evidence of an effect, or evidence of no-effect. I also really like the idea of the game and would be very interested to hear how this will be disseminated later on. I do have some comments however and some details of the study proposal are not clear to me: In the discussion you rightly state that any new intervention for SAM should focus on the time of in-patient treatment when mortality is highest and you mention "1-2 weeks after admission". It is known from literature that mortality is highest in the first 48 hours. I do appreciate and understand your reasoning that for this phase 2 trial you will only enrol children who are clinically stable (i.e. at the moment of the start of the Transition phase) but this will definitely cause enrolment/recruitment bias favouring the children that have made it to the Transition phase in the first place and that have survived the stabilisation phase. You need to mention this important consideration in the background as well since this is an important decision in the design of the trial.
--

	Background: P6: do provide a more recent reference on SAM mortality than the Amadi et al paper from 2001 please. On page 7 you end the 2nd paragraph stating that the damage to the small intestinal mucosa plays an important role in the 'genesis' of.....adverse nutritional status". Can you be sure it is the cause and not the consequence? P8: what is the rationale behind the "within 72 hours" of initiation of transition? Does this have to do with the Zambian 2-3 days of being able to provide informed consent (as mentioned on page 14?) I do miss why an important marker for intestinal inflammation, faecal calprotectin, is not chosen as an outcome parameter (and not even mentioned in the proposal). As it is used in high-income settings on a daily basis in paediatric inflammatory GI disorders, the authors should comment on this choice. I would describe the trial 'single blinded' as the study personnel examining the outcome (i.e. in the lab) are blinded to the intervention. I need more clarity and some explanation on the choice for a composite variable rather than using the 3 individual biomarkers (Myeloperoxidase, neopterin and AAT). What is the difference (p12) between 'equation 1' (EE score) and 'equation 2' (EE biomarker score)? On page 13 it is referred to as 'enteropathy activity score'; this is confusing. Please elaborate. I miss MUAC as an important 'core covariate' (p13); explain please. As there will be very many AE's, it is important to define an adverse event. Do explain how feasible it will be in daily clinical practice to 'review all AE's for causality, expectedness and severity'. Our group has chosen in the recent past to only focus on SAE's since AE's are inherent to SAM treatment. Explain please.
--	--

REVIEWER	Kirk Tickell University of Washington
REVIEW RETURNED	03-Jan-2019

GENERAL COMMENTS	I would like to commend the authors of this study. It is very pleasing to read such an innovative SAM study. The protocol is easy to read and should be accepted for publication after a few changes. 1) I cannot find the study's trial registry entry. While BMJ open is a great venue to offer a detailed, digestible protocol, it would still be helpful to have a link included in the manuscript that goes to the trial registry. 2) It may be beneficial to have the abstract of this manuscript be clear that this is a pilot study (or group of pilots). It is mentioned later in the manuscript, but being up front about it now may benefit the authors in the future.
--

	3) Defining “clinically stable” is always going to be a sticking point. The definition on page 9 line 3-5 is helpful. The authors might consider adding not requiring supplementary oxygen to the definition, and possibly not requiring NG feeding. I also wonder how they would handle a rehydrated child who still has diarrhea, particularly given that some their interventions have potentially osmotic effects. 4) The retention of caregivers in hospital may be problematic, but it is an understandable choice. Two comments related to that: a. Are the beds free, and is care free at all sites? If not, will you cover cost of hospitalization or the costs of additional days? If it is free, are you providing a travel/food stipend for the families? b. Sites that retain SAM children beyond stabilization often have higher rates of absconding/leaving against advice. You have accounted for this in the power calculation, but how will it be handled in the analysis? Is there an avenue to get participants to return to a research clinic for dosing, or if that is too far for them to travel regularly, could they come back for the final test? 5) The analysis section should state clearly whether this will be an intention to treat or as-per-protocol analysis. 6) This group has extensive experience collecting stool, but will you apply any inclusion/exclusion criteria about stool collection? If a child has no baseline stool collected on day 1 & 2 will they be included? 7) Is there evidence that Bovine collustrum contains or may promote production any of the inflammatory markers you will be measuring for your primary outcome?. The AJTMH MALED/Kosek paper (Am. J. Trop. Med. Hyg., 96(2), 2017, pp. 465–472) noted that breast feeding was associated with increased inflammatory markers, which may or may not reflect the true state of the gut. Breast feeding may be a useful covariate for your analysis. 8) Page 10, line 28 – implies that the Mesalazine result gives a blank endorsement of immuno-modulation as safe in this population, rather than proof that Mesalazine and its mechanism of action are safe. The following sentences do clarify the importance of also examining budesonides safety, but I might consider tweaking that Melsalazine sentence a little, perhaps “some immune-modulatory methods are safe.” Given the authors immunology expertise I would be completely happy if they disagree with me. 9) It wasn’t clear to me if sites have a recruitment cap. Will you stop or slow recruitment if one site is racing in front of the others? 10) Have you considered standardizing the transition therapeutic food to either RUTF or F-100? It may becoming an unnecessary confounder if there is a stockout. In a similar vein, will you collect data on pre-enrollment management, particularly days of antibiotics and type of antibiotics. We don’t know a lot about how antibiotics effect EE markers at the moment, so while exposure to antibiotics will be universal in your population it may be worth capturing which antibiotics they got and for how long.
--	---

	11) The protocol states enrollment happens at the point of transition, but also that therapies will be started on day 2 of transition. The authors should clarify when randomization will happen. If it happens at enrollment, they need to clarify how events occurring after randomization but before treatment initiation will be handled, are they considered SAEs, will children who died after randomization but before treatment initiation be replaced. This is less of an issue if randomization occurs immediately prior to the therapies starting. 12) I was surprised the authors are committing to so many covariates to include in their analysis when some of these factors may be balance by randomization (sex) and others may occur in only a small number of participants (HIV ~10%?). Potentially you'll have 72-85 children in each model and you're committing to include 8 covariates. You may have decided that you think these variables will add precision to your models, but still think it would be appropriate to leave some wiggle room to drop factors like sex, oedema, HIV, and diarrhoea if they are either balanced at baseline or they are too rare to meaningfully include in a model. Additionally, I imagine you may wish to display a crude model and draw inference from this too, so you should consider adding it to the analysis plan. 13) Finally, is there a need for this study to report to the FDA or other licensing body for any of these products? Could WHO endorse these all of these products without approval or changes to their licensing status after the phase 3 study. It might be helpful to add information about the approval status to the manuscript as most readers will not be familiar with these products (I wasn't).
--	---

VERSION 1 – AUTHOR RESPONSE

Reviewer 1

1 We have provided a fuller discussion of the risks and benefits of teduglutide. The literature relating to children is included under Interventions (p7-8) and discussed more fully on p14.

Reviewer 2

1 This is an important point. As this is the first use of these agents for this disorder, we elected to delay recruitment until the children are clinically stable. We anticipate that AEs will be identified more clearly at or after transition when the child is more stable. The reviewer is absolutely correct to point this out and we have included this in the Discussion (page 13).

2 We have referenced additional papers on SAM mortality at this point (reference 12 has been moved up from 21, and ref 13 added).

3 We have added a comment to the effect that malnutrition may also cause gut damage (p5).

4 Yes, we allowed up to 72 hours from transition to allow this important process to take place (p12).

5 We elected to use three faecal biomarkers as there is considerable experience with them in the Mal-ED study. We have added a comment about calprotectin (p13).

6 Single blinded is an appropriate term; we have included it (abstract).

7 The use of a composite marker is based on the Mal-ED data (reference 34). We elected to use a composite marker as we need a single measure for use as a primary endpoint. In the original paper by Kosek et al, the faecal biomarkers were categorised. In the TAME trial we felt that this would lead to loss of information (see refs 35 and 36), so we decided to modify the equation so that, while

retaining the composite score and weighting, we can use all data as continuous variables. We have highlighted the justification for this on p9-10.

8 MUAC is included as an inclusion criterion (as part of the definition of SAM), but the reviewer is right that we could analyse it too (though distribution of this variable will be restricted) now included in Table 2.

9 We feel the need to capture as many AEs as possible as these are novel therapies for this disorder. We are painfully aware that this will not be easy!

Reviewer 3

1 We have included a link to the trial registry (<https://clinicaltrials.gov/ct2/show/NCT03716115>).

2 We described this as a phase 2 trial; we have made this more explicit in the abstract.

3 We have added supplementary oxygen to the definition (p6), but as many of the children will have ongoing diarrhoea (indeed they may be most likely to benefit) we should not exclude NG tubes.

4 The retention of caregivers in hospital will only be for a few days, and we have done this before. However, we appreciate that this brings problems. Bed-days are free, so there will be no charges incurred. We will do our best to encourage caregivers to bring the children back in the event of early discharge.

5 This will be a per-protocol analysis; we have stated this on p10.

6 Yes, we will have to exclude children who cannot provide a stool sample as this is a primary endpoint. We have included this in the analysis section (p11).

7 We anticipate that colostrum will reduce stool inflammatory markers, and the Mal-ED results are hard to interpret. It should be noted that we will also have some biopsy data to help with interpretation of stool biomarkers. Breast feeding will be a covariate in the analysis (p10).

8 We have softened the statement on safety of immunomodulatory agents (p8).

9 No recruitment cap is anticipated.

10 We follow standard protocols, with individualised decisions on the choice of F100 or RUTF. This will be noted, but we expect randomisation to ensure even allocation to treatment groups. The point about antibiotics is well taken; the CRFs are designed to record antibiotic use in some detail.

11 We hope that enrolment, randomisation and initiation of treatment will occur quickly, as the most significant delay will be time taken for caregivers to decide on consent. Once randomised, treatment initiation will be immediate.

12 The list of covariates is now 8, once we have included breastfeeding.

13 It would be premature to report to FDA the results of a phase 2 trial. The point is an important one, though, and certainly if any of these therapies make it to a phase 3 trial this would have to be considered carefully.

VERSION 2 – REVIEW

REVIEWER	Wieger Voskuijl Amsterdam University Medical Centre, Emma Children's Hospital, Amsterdam, The Netherlands
REVIEW RETURNED	22-Feb-2019

GENERAL COMMENTS	Dear Editor and Authors, I have seen and read the amendments to the protocol and I am happy with them.
--

REVIEWER	Kirk tickell University of Washington
REVIEW RETURNED	04-Feb-2019

GENERAL COMMENTS	Thank you for addressing/answering those items. Good luck!
--